# Protocol for a scoping review of implementation research approaches to universal health coverage in Africa

Chukwudi A Nnaji ![ORCID],[1,2] Charles S Wiysonge ![ORCID],[1,2,3] Joseph Okeibunor ![ORCID],[4] Thobile Malinga ![ORCID],[2] Abdu A Adamu ![ORCID],[2,3] Prosper Tumusiime ![ORCID],[4] Humphrey Karamagi ![ORCID] [4]

► Prepublication history and additional materials for this paper is available online. To view these files, please visit the journal online (http://dx.doi.org/10.1136/bmjopen-2020-041721).

[1]School of Public Health and Family Medicine, University of Cape Town, Cape Town, South Africa
[2]Cochrane South Africa, South African Medical Research Council, Cape Town, South Africa
[3]Department of Global Health, Faculty of Medicine and Health Sciences, Stellenbosch University, Stellenbosch, South Africa
[4]Health Systems and Services Cluster, World Health Organization Regional Office for Africa, Brazzaville, Congo

**Correspondence to**
Dr Chukwudi A Nnaji;
nnjchu001@myuct.ac.za

## ABSTRACT

**Introduction** Implementation research has emerged as part of evidence-based decision-making efforts to plug current gaps in the translation of research evidence into health policy and practice. While there has been a growing number of institutions and initiatives promoting the uptake of implementation research in Africa, their role and effectiveness remain unclear, particularly in the context of universal health coverage (UHC). This review aims to extensively identify and characterise the nature, facilitators and barriers to the use of implementation research for assessing or evaluating UHC-related interventions or programmes in Africa.

**Methods and analysis** This scoping review will be developed based on the methodological framework proposed by Arksey and O'Malley and enhanced by the Joanna Briggs Institute. It will be reported in accordance with the Preferred Reporting Items for Systematic Reviews and Meta-Analyses extension for Scoping Reviews guidelines. A comprehensive search of the following electronic databases will be conducted: Medline (via PubMed), Scopus and the Cochrane Library. Relevant grey literature and reference lists will also be searched. All publications describing the application of implementation research in the context of UHC will be considered for inclusion. Findings will be narratively synthesised and analysed using a predefined conceptual framework. Where applicable, quantitative evidence will be aggregated using summary statistics. There will be consultation of stakeholders, including UHC-oriented health professionals, programme managers, implementation researchers and policy-makers; to provide methodological, conceptual and practical insights.

**Ethics and dissemination** The data used in this review will be sourced from publicly available literature; hence, this study will not require ethical approval. Findings and recommendations will be disseminated to reach a diverse audience, including UHC advocates, implementation researchers and key health system stakeholders within the African region. Additionally, findings will be disseminated through an open-access publication in a relevant peer-reviewed journal.

## Strengths and limitations of this study

► This scoping review will be conducted in accordance with an enhanced evidence synthesis methodology.
► It will use a well-grounded conceptual framework to map the evidence on implementation research in the context of universal health coverage.
► Multiple databases will be searched with a comprehensive search strategy to identify both peer-reviewed and relevant grey literature sources.
► Broad consultation with stakeholders will be incorporated to enhance the review's conceptual and methodological rigour.
► Due to the broad nature of the topic, it is possible that some relevant literature may not be identified by our search strategy, however comprehensive.

identified as a vital step for achieving universal health coverage (UHC) and equitable access to quality healthcare.[1 2] It has been recognised that decisions informed by research evidence have the potential to promote equitable access to health services and improve health outcomes at the population level, while strengthening health systems.[2] The WHO defines UHC as 'ensuring that all people have access to needed health services (including prevention, promotion, treatment, rehabilitation and palliation) of sufficient quality to be effective while also ensuring that the use of these services does not expose the user the financial hardship'.[3] Since the 1978 Alma-Ata Declaration and the 1986 Ottawa Charter for Health Promotion, the right to the highest attainable standard of physical and mental health has gained increasing attention.[4] As a result of this prioritisation, UHC was adopted as a target of the Sustainable Development Goals, with the aspiration that countries will achieve this by 2030.[5]

With the increasing momentum of global efforts towards the attainment of UHC, countries are often faced with difficult choices

## INTRODUCTION

The need for health decision-making to be informed by empirical evidence has been

regarding the most effective use of available health resources, particularly in contexts of resource limitation, competing healthcare needs and political priorities.[6] Given this inherent complexity, UHC decision-making requires adequate consideration of best available and contextually applicable research evidence.[6 7] While investment in health research and research outputs have grown considerably in Africa over the years, there remain enormous gaps in translating available research evidence into health policy and practice.[8] This so-called 'know-do gap' has resulted in suboptimal gains from allocated health resources, in spite of growing investment towards the actualisation of UHC in Africa.[2 9] The gap is accentuated by the region's high burden of communicable and non-communicable diseases.[10 11]

Implementation science has emerged in response to this critical gap.[12] Implementation science is an integral part of the broader evidence-informed decision-making (EIDM) enterprise. EIDM involves processes of distilling and disseminating the best available evidence from research, practice and experience and using that evidence to inform and improve public health policy and practice.[13 14] Knowledge translation, knowledge transfer and translational research are EIDM concepts that are closely related to implementation science, used to refer to the processes of moving research-based evidence into policy and practice, through the synthesis, dissemination, exchange and application of knowledge to improve the health of the population.[13 15–17] Although there may be nuanced differences in their conceptualisation, these terms essentially have similar goals and practical implications for improving health outconmes.[15–17]

There has been no clear consensus on the definition of implementation science.[18] In 2015, Odeny *et al* published a review of the literature that found 73 unique definitions.[19] Broadly, implementation science has been defined as: 'the scientific study of methods to promote the systematic uptake of research findings and other evidence-based practices into routine practice, and, hence, to improve the quality and effectiveness of health services'.[16] Since the field of implementation science has cogent applications in both clinical and public health settings, this definition is more encompassing and underscores the field's broad nature. The process of inquiry in implementation science is through research, which builds on traditional scientific methods, but focuses on a unique set of questions to improve the use of research in implementation.[16 19] Thus, implementation science offers the toolkit for addressing the know-do gap.[16 20 21]

Implementation research is an emerging subdomain of implementation science that has been more distinctively defined. In 2006, Eccles and Mittman proposed a working definition for the field of implementation research—defining it as the 'scientific study of methods to promote the adoption and integration of evidence based practices, interventions and policies into routine healthcare and public health settings.'[21] More recently in 2013, the WHO's Alliance for Health Policy and Systems Research (AHPSR) defines it as 'the scientific study of the processes used in the implementation of initiatives as well as the contextual factors that affect these processes.'[18] This definition highlights a defining feature of implementation research; that is, going beyond the study of methods of promoting the uptake of evidence into routine practice, to studying the contextual facilitators and barriers to evidence-based implementation.[17 18] For this reason, implementation research has been regarded as the heart and soul of implementation science.[17] While implementation science and implementation research have been interchangeably used in literature, implementation research will be the reference term for this review.

Various conceptual theories and frameworks have been used to guide implementation research efforts across diverse settings. Some of the most commonly used frameworks include the Consolidated Framework for Implementation Research, Theoretical Domains Frameworks, diffusion of innovations, reach effectiveness adoption implementation maintenance, Quality Implementation Framework, Interactive Systems Framework and normalisation process model.[22 23] Additionally, the use of adapted forms or combination of these frameworks has been reported.[22] To facilitate the use of implementation research in health system decision-making and routine practice, there have to be: (a) availability of rigorous, robust, relevant and reliable evidence, (b) decision-makers' appreciation of the value and importance of empirical evidence in decision-making processes and (c) a trusting, mutually respectful and enduring engagement between evidence producers and decision-makers.[6 13 24]

Various implementation research initiatives and efforts for evaluating and improving health programme outcomes have emerged in the African region in the last decade.[13 17 25–28] In spite of this substantial growth, implementation research uptake, effectiveness and scale-up in the region are challenged by numerous barriers.[25–27] These include inadequate research funding, limited availability and access to good quality research and paucity of contextually relevant evidence.[27] Other reported barriers include the untimeliness of research output and, of course, fragile collaboration between researchers and users of evidence like policy-makers and frontline programme implementers.[2 7 29 30]

## Study rationale

Globally, evidence-based health decision-making and implementation models are being adopted as approaches for improving the health of populations.[7 16 31] While there has been a growing number of institutions and initiatives promoting the uptake of implementation research in Africa, the role and effectiveness of these initiatives remain unclear, particularly in UHC contexts.[32 33]

Synthesised bodies of evidence on the role of implementation research in Africa's health systems and the extent to which it has been used to promote UHC and health equity on the continent are sparse. With limited funding and institutional research capacity to drive

implementation research efforts in Africa, there is an urgent need to seek out cross-country learning opportunities that can bolster understanding of implementation research and broader EIDM strategies in the region.[11 34] A better understanding will be important to stimulate greater uptake, better application and sustainability of implementation research strategies within UHC contexts in the region.

Scoping reviews represent an appropriate methodology for thematically reviewing large bodies of literature in order to generate an overview of existing knowledge and practice, as well as identifying existing evidence gaps.[35 36] Like full systematic reviews, scoping reviews employ methods that are transparent and reproducible, using predefined search strategies and inclusion criteria.[37 38] However, unlike systematic reviews which often target specific and narrow research questions, scoping reviews typically have a broader focus—including the nature, volume and characteristics of the literature in order to identify, describe and categorise available evidence on the topic of interest.[36–38] This scoping review will be valuable for filling existing gaps in the availability of synthesised evidence on implementation research in the context of UHC, health equity and health systems strengthening within the African region. Additionally, it will map the region's implementation strategies, major actors, reported outcomes, facilitators and barriers from a diverse body of literature. Ultimately, it seeks to provide a holistic and user-friendly evidence summary of implementation research and key issues in the region for researchers, policy-makers and implementers, while identifying lingering knowledge and practice gaps to inform future implementation research efforts.

### Study objectives
The aim of this review is to extensively scope the literature to identify and characterise the nature, facilitators and barriers to the use of implementation research for assessing or evaluating UHC-related interventions or programmes in the African region.

## METHODS
### Conceptual framework
This scoping review will follow the implementation science taxonomy proposed by Ridde *et al*[39] to guide the synthesis of identified evidence and characterising the nature of implementation research strategies in the context of UHC. To help characterise and describe the possible implementation research approaches, frameworks and theories, this taxonomy defines four models commonly used in implementation science (intervention theory, framework, middle-range theory and grand theory). These models form a continuum and may overlap when used. Implementation scientists and researchers use these models for three main implementation studies: fidelity assessment, process evaluation and complex evaluation.[39]

### Study design
The design of this scoping review will be developed based on the Arksey and O'Malley scoping review methodology,[40] as enhanced by the Joanna Briggs Institute (JBI).[41] The JBI's enhanced framework expands the six stages of Arksey and O'Malley into nine distinct stages for undertaking a scoping review: (1) defining the research question; (2) developing the inclusion and exclusion criteria; (3) describing the search strategy; (4) searching for the evidence; (5) selecting the evidence; (6) extracting the evidence; (7) charting the evidence; (8) summarising and reporting the evidence and (9) consulting with relevant stakeholders.

### Stage 1: defining the research question
Through consultation with the research team and key stakeholders, the overall main research question was defined as: 'What are the nature, facilitators and barriers of implementation research strategies for promoting UHC in Africa?' For the purpose of this review, implementation research has been defined within the broader frameworks of implementation science, knowledge translation and evidence informed decision-making. Based on the primary research question, the following specific research questions were defined:
1. How has implementation research been used to assess or evaluate UHC-related interventions or programmes in the African Region?
2. What are the contextual facilitators and barriers to the application of implementation research in assessing or evaluating UHC-related interventions or programmes in Africa?

### Stage 2: developing the inclusion and exclusion criteria
#### Inclusion criteria
These will be defined based on the population, concept and contexts framework, proposed by Peters *et al*.[42] This framework is more appropriate for scoping reviews, compared with the commonly used population, intervention, comparator and outcome (PICO) framework, as it allows for the consideration of publications that may not feature all of the four PICO elements (eg, lacking an outcome or comparator/control). Eligible population will include evidence producers (health researchers), intermediaries (such as knowledge brokers and implementation research institutions) and evidence users (such as health policymakers, programme implementers like nongovernment organisations and healthcare providers). The concept of interest is implementation research. To be considered for inclusion, studies must report on the use of implementation research strategies, models, theories or frameworks for assessing or evaluating UHC-related interventions or programmes. These may include activities such as fidelity assessment, process evaluation, outcome evaluation or complex evaluation.[39] Studies with or without comparison between implementation research strategies and controls will be eligible for inclusion. UHC outcomes will include scope of package of health services;

population coverage, access and service utilisation; quality of care; and financial risk protection, in line with the Cube framework.[43] Studies that evaluated specific health programme implementation outcomes, barriers or facilitators, will be included, provided the evaluation involved the use of specified implementation research approaches, frameworks or models. Context will be health systems within the African region (online supplemental appendix 1 specifies the countries and territories of focus within the region). Any type of primary study design will be eligible, including randomised controlled trials and observational studies.

### Exclusion criteria

Literature focused solely or mainly on theoretical and conceptual development of implementation research will be excluded; as will study protocols and studies evaluating implementation outcomes without specifying or mentioning the implementation research approaches, models, theories or frameworks used. Multinational literature involving African and non-African countries and meeting inclusion criteria will be excluded if country-specific information cannot be abstracted.

### Stage 3: describing the search strategy

The search strategy will be developed with the guidance of a reference librarian, and adapted for other databases using appropriate controlled vocabulary and syntax. The search strategy will use search terms that are sensitive enough to capture literature sources relevant to implementation research, with due cognisance of the field's diverse and overlapping nomenclature and search filters for African countries. An initial exploration of current available literature on implementation research and UHC will help guide the selection of search terms, ensuring they are inclusive enough to capture any UHC-related implementation research intervention. The search strategy will be applied in accordance with the Peer Review of Electronic Search Strategies guidelines.[44] A provisional Medline search strategy is illustrated in online supplemental appendix 1.

### Stage 4: searching the evidence

A comprehensive literature search will be conducted on the following electronic databases: Medline (via PubMed), Scopus and Cochrane Library (including the Cochrane Central Register of Controlled Trials and the Database of Abstracts of Reviews of Effects). Each database will be searched from the year 2000 (coinciding with the inception of implementation science as a field in the mid-2000s) to the date of search. Additionally, relevant grey literature will be searched for implementation research-related reports, including the website of the WHO AHPSR. Websites of known implementation research institutions, networks and collaborations will be explored. We will also conduct hand-searches of reference lists of relevant literature to identify potentially

eligible literature. Only literature sources published in English will be eligible for inclusion.

### Stage 5: selecting the evidence

The review process will consist of two levels of screening: a title and abstract screening to identify potentially eligible publications and review of full texts to select those to be included in the review based on predefined inclusion/exclusion criteria. The first level will involve the independent screening of titles and abstracts of all retrieved citations from the search output by CAN and TM. Articles that are deemed relevant will be included in the full-text review. Following the removal of duplicates, full texts of remaining studies will be retrieved. In the second step, the retrieved full texts will be assessed in duplicate by CAN and TM to determine if they meet the inclusion/exclusion criteria. Those meeting the inclusion criteria will be included in the review. Discrepancies in study selection between the two independent reviewer will be discussed to reach a consensus. Where a consensus is not reached, a third reviewer (CSW) will arbitrate.

### Stage 6: extracting the evidence

A data extraction tool (using a Microsoft Excel spreadsheet) will be developed by the research team to extract relevant info from included literature. Information to be extracted will include at least the following:

1. Author(s).
2. Year of publication.
3. Country where the evidence/study was published/conducted.
4. Aims/purpose.
5. Study population and study size.
6. Type of evidence/study design.
7. Implementation research strategy, model, theory or framework used.
8. Duration of implementation research.
9. Type of UHC-related programme or intervention involved (classified by programmatic area of focus).
10. Key implementation research findings.
11. Reported implementation research facilitators and barriers.

Other categories that come up during the data extraction process will be discussed by the research team and added to the data extraction tool. The tool will be reviewed by the research team and pretested before use. Data abstraction will be conducted in duplicate by two independent reviewers. To ensure accurate data collection, each reviewer's independently abstracted data will be compared, and any discordance will be resolved through a consensus. Where a consensus is not reached after discussion between the two independent reviewers, a third reviewer will arbitrate. All collected data will be collated in a single Microsoft Excel spreadsheet for validation and coding.

## Stage 7: charting the evidence

A table describing each included study will be presented using the 11 information headings described in stage 5 above. To ensure accuracy of charted evidence, each reviewer's independent charted data will be compared and any discrepancies will be iteratively discussed by the researchers to ensure consistency between the reviewers.

## Stage 8: summarising and reporting the evidence

Findings of the review will be reported using the Preferred Reporting Items for Systematic Reviews and Meta-Analyses (PRISMA) extension for Scoping Reviews check-list.[45] A PRISMA flow diagram will be used to illustrate the literature search results and study selection process. Findings will be summarised and reported using narrative descriptions based on the following themes: country-context, implementation research strategy used and type of UHC-related programme or intervention involved. The implementation science taxonomy proposed by Ridde *et al*[39] will be used to classify identified implementation research models, theories or frameworks. Implementation research facilitators and barriers will be reported based on the themes that will emerge from the charted evidence. Where applicable, quantitative evidence will be aggregated using summary statistics. As the purpose of a scoping review is to aggregate evidence and present a summary of the evidence rather than to evaluate the quality of the individual evidence, this review will not involve any formal appraisal of the quality of included evidence.

## Stage 9: consultation

Multidisciplinary and multinational consultations will provide opportunities for stakeholders to provide additional insights beyond what is reported in the literature.[46] Given the potentially diverse nature of implementation research literature, a broad array of stakeholders will be consulted, from implementation researchers to UHC-oriented health professionals, programme managers and policy-makers. These stakeholders can help to identify grey literature that may not be obtainable from scholarly database searches, as well as providing methodological, conceptual and practical insights for guiding the interpretation and dissemination of findings.

## Patient and public involvement

Patients and the public were not involved in the development of this protocol.

## ETHICS AND DISSEMINATION

Since the scoping review methodology involves reviewing and collecting data from publicly available materials, this study will not require ethics approval. To facilitate dissemination of findings, the research team will use a multistakeholder approach in presenting the findings to key health system stakeholders within the African region, in addition to open-access publication in a relevant peer-reviewed journal.

**Acknowledgements** The authors gratefully acknowledge the valuable comments and inputs from the following colleagues at Cochrane South Africa: Chinwe Iwu-Jaja, Selvan Naidoo, Phelele Njenje, Jill Ryan, and Alison Wiyeh.

**Collaborators** Not applicable.

**Contributors** The study was conceived by CSW, JO and HK. CAN wrote the first draft of the manuscript with guidance from CSW. CSW, JO, TM, AAA, PT and HK contributed to writing the final version of the manuscript. All the authors read and approved the final manuscript.

**Funding** The WHO Regional Office for Africa and the South African Medical Research Council (SAMRC), through Cochrane South Africa, provided funding for this study.

**Disclaimer** The views expressed are those of the authors and do not necessarily reflect those of WHO, the SAMRC, Cochrane, or any other organisation that the authors are affiliated with.

**Competing interests** None declared.

**Patient consent for publication** Not required.

**Provenance and peer review** Not commissioned; externally peer reviewed.

**ORCID iDs**
Chukwudi A Nnaji http://orcid.org/0000-0002-4132-1922
Charles S Wiysonge http://orcid.org/0000-0002-1273-4779
Joseph Okeibunor http://orcid.org/0000-0002-6696-8503
Thobile Malinga http://orcid.org/0000-0001-5512-6696
Abdu A Adamu http://orcid.org/0000-0003-3317-1319
Prosper Tumusiime http://orcid.org/0000-0001-6899-824X
Humphrey Karamagi http://orcid.org/0000-0002-6277-2095

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
