## [Reviewer comments · BMJ Open]

ARTICLE DETAILS

TITLE (PROVISIONAL)	A protocol for a scoping review of implementation research approaches to universal health coverage in Africa
AUTHORS	Nnaji, Chukwudi; Wiysonge, Charles; Okeibunor, J; Malinga, Thobile; Adamu, Abdu; Tumusiime, Prosper; Karamagi, Humphrey

VERSION 1 – REVIEW

REVIEWER	Lara Gautier School of Public Health, University of Montreal
REVIEW RETURNED	03-Aug-2020

GENERAL COMMENTS	This is an interesting scoping review protocol. Not sure why the authors did not simply submit it to Protocols.io (would have been enough considering the foreseeable impact of such scoping review), especially given that they only report on early steps taken in the research process (e.g., the full list of keywords and tentative search strategies are not provided). 1. Major comments My main concern is the lack of focus of the research objective. The authors are trying to do too many things at the same time. For a scoping review (SR) to have coherence, they would need to concentrate on one single and clearer objective. There are also a number of caveats in this SR protocol which would impede publication. Specific comments: *Research question and objective are not clear. Please ensure coherence between abstract, highlights, introduction and step 1. In step 1 the authors do not even specifically refer to UHC. The specific RQs take very different directions. I suggest emphasising on question 4 (i.e., the question linked to knowledge translation), which is IMO the only one that the authors can truly address through a scoping review. *Also, not clear why you would assign performative goals to IR (e.g., ensuring quality care and population coverage). Please provide a rationale for making this choice. *Relatedly, the choice of the UHC cube (which highlights UHC outcomes) as conceptual framework for this SR is debatable, and sounds counter-intuitive given the focus on IR papers. *Please consider new foundational publications on IR applied to the field global health and published in BMJ journals (e.g.: https://gh.bmj.com/content/5/4/e002269) and elsewhere. *The section on the two "concepts of interest" reads unclear. What do the authors mean by "intervention concept" and "outcome concept"? How could IR be considered an "intervention concept"? This needs to be revised.
--

	*If your Context is the whole African continent, you should at least mention this instead of stating "Africa" broadly, and also specify the nb of countries of "Africa" you will be investigating. Also, please note that the Canary Islands are part of Spain... *Languages. In appendix 1, your search terms are exclusively provided in English. Why do you state that there will be no restrictions of language? By looking only into these common English-language databases, you will de facto primarily find publications in English language. Also, if you really aim to reach this ambitious goal, you will need to a) look into each non-English publication databases (and provide their names in the current protocol) and b) make sure to provide details on translation issues (what translating tools will be used? what quality checks?, etc). *Exclusion criteria: on what basis would you remove items "evaluating implementation outcomes using specific IR approaches"? *Unclear whether the authors will perform a quality assessment of the selected papers. *Appendix 1: why would you look for "evidence-based medicine" papers? This is not coherent with your research focus. There are also a sheer number of keywords that are missing from this provisional search strategy. Please refer to existing published systematic and scoping reviews on the topic of UHC. *Lastly, many tools are now freely available to researchers who perform literature reviews. I would strongly recommend the authors to use a common review extraction tool, such as Rayyan (https://rayyan.qcri.org/welcome) as it would ensure a more transparent and rigorous item screening and selection than Excel. 2. Minor comments *Please remove the sections "Ethics and dissemination" and "Patient and public involvement" as these are not applicable to a scoping review protocol. *Unclear why the PRISMA guidelines are provided at the end of the submission - especially since the authors do not apply the PRISMA-recommended PICO strategy. *While the level of English language is acceptable in general, I would suggest a thorough English proofreading done by a professional language editor.
--	--

REVIEWER	David Watkins University of Washington, USA
REVIEW RETURNED	25-Aug-2020

GENERAL COMMENTS	This scoping review protocol by Nnaji and colleagues addresses an applied-research question of great importance to health policy and practice in Africa. The review will attempt to identify and characterize health implementation research initiatives in African countries. I do not have any major concerns with the protocol, but I offer a few comments and suggestions for the authors' consideration: p10 (line 27): some experts think that the UHC cube does not adequately address quality of care. In this conceptual framework (and in the study inclusion/exclusion criteria and charting approach), I think the authors could distinguish between "population coverage" (the name of the dimension on the UHC cube), which is mostly about aggregate access, and quality, which is a modifier of individual health outcomes. My read of the literature suggests that most implementation research studies tend
---

	to focus on either improving access (expanding coverage to marginalized clients) or improving the quality of care received by those already in the system. Of course, there are some overlaps. Ng and colleagues (PLoS Medicine, 2014) talk about "effective coverage" as the combination of access (utilization) and quality, and I think it would be helpful to use this term or at least to disentangle quality improvement from population coverage. p11 (stage 1): in light of the above, I might suggest re-wording or re-structuring the research objectives a bit. The 4 research questions do not align well with the conceptual framework (UHC cube); they introduce new constructs (eg, resilience/sustainability; barriers to uptake of research) and do not speak at all to two of the three UHC cube dimensions: financial protection and the scope of the UHC benefits package (ie, services covered under UHC). p12 (inclusion criteria): again, I suggest being more clear about the UHC outcomes of interest. Lines 20-24 mention health service coverage (not sure if this refers to population coverage or scope of benefits package), access (including utilization and quality together - see above comments), and financial protection... but not much about resiliency, evidence-to-policy, etc. I suggest a modification of Figure 1 that shows the relationships between the study objectives and the dimensions of the cube and the outcomes of interest, to make the whole strategy feel more coherent. p12 (exclusion criteria): the authors say they will exclude "outcomes that are not UHC-related." Some additional explanation is warranted here. If the scope of this review is only studies that look at general health systems strengthening to improve overall access, financial protection, etc, then I think the search will turn up with very little. However, there could be a wide range of disease- or program-specific studies that speak to UHC outcomes like access and financial protection without explicitly using the term UHC or looking at the whole health system. For example, a paper may use a quasi-experimental approach (and robust implementation science methods) to evaluate the impact of a new HIV care delivery model on ART uptake and out-of-pocket costs. What to do about this sort of study? It might be useful for the authors to cite a few exemplar studies (eg, at the end of this section at the top of p13) to give the reader a better sense for what sort of implementation research studies the authors hope to pick up (or not) with the review. p13 (line 30): minor point: why search databases "from inception"? The modern UHC movement only goes back to about 2005, and the field of implementation research is not much older. I worry about clouding the review with a bunch of studies of vertical programs from the 1980s/90s that have less relevance in today's time. p14 (line 37): another minor point: I might explicitly include protocols for research studies in the inclusion criteria and in the extraction/charting process. I would guess that there are a number of large-scale multi-year studies that are ongoing but not published (except in protocol form). p15 (stage 7): the authors could consider providing a few examples of how they intend to visualize and interpret the charted
--	--

	evidence. Maybe mention some relevant disaggregations or ways of summarizing the evidence? p16 (lines 7-9): I wonder whether stakeholders should be involved in the identification of grey literature at the study identification phase (lit search) rather than at the dissemination phase? Given the broad objective of this research, it may be worthwhile to have a multidisciplinary, multinational advisory committee for this work that could be engaged from beginning to end.
--	--

VERSION 1 – AUTHOR RESPONSE

Reviewer 1

Comment: This is an interesting scoping review protocol. Not sure why the authors did not simply submit it to Protocols.io (would have been enough considering the foreseeable impact of such scoping review), especially given that they only report on early steps taken in the research process (e.g., the full list of keywords and tentative search strategies are not provided).

Response: We thank the reviewer for affirming the importance and potential impact of our work. We also appreciate the suggestion of Protocols.io. We will consider submitting future protocols of this nature to it.

Comment: My main concern is the lack of focus of the research objective. The authors are trying to do too many things at the same time. For a scoping review (SR) to have coherence, they would need to concentrate on one single and clearer objective. There are also a number of caveats in this SR protocol which would impede publication.

Response: We appreciate this valuable comment. We have refined and clarified the study objectives/research questions (lines 158 – 159 and 188 – 190)

Comment: Research question and objective are not clear. Please ensure coherence between abstract, highlights, introduction and step 1. In step 1 the authors do not even specifically refer to UHC. The specific RQs take very different directions. I suggest emphasising on question 4 (i.e., the question linked to knowledge translation), which is IMO the only one that the authors can truly address through a scoping review.

Response: We have revised the research questions accordingly in the abstract (lines 19 – 20), objectives section (lines 158 – 159) and under Step 1 (lines 182 – 190).

Comment: Also, not clear why you would assign performative goals to IR (e.g., ensuring quality care and population coverage). Please provide a rationale for making this choice.

Relatedly, the choice of the UHC cube (which highlights UHC outcomes) as conceptual framework for this SR is debatable, and sounds counter-intuitive given the focus on IR papers.

*Please consider new foundational publications on IR applied to the field global health and published in BMJ journals (e.g.:

<https://eur01.safelinks.protection.outlook.com/?url=https%3A%2F%2Fgh.bmj.com%2Fcontent%2F5%2F4%2Fe002269&data=04%7C01%7Cnnjchu001%40myuct.ac.za%7C23c12ff9896942b5002008d875b89d40%7C92454335564e4ccfb0b024445b8c03f7%7C0%7C0%7C637388783062272068%7CUnknown%7CTWFpbGZsb3d8eyJWljoiMC4wLjAwMDAiLCJQIjoiV2luMzliLCJBTiI6IjEhaWwiLCJXVCi6Mn0%3D%7C1000&sdata=yE1DERc7FOQ5GoDvHvBgd0aPMwpORFIaMIBNo1qQWmU%3D&reserved=0>) and elsewhere.

Response: We thank the reviewer for the this insightful suggestion. We have revised the review's conceptual framework and have incorporated the implementation science taxonomy proposed by Ridde and colleagues (Lines 163 – 170).

Comment: The section on the two "concepts of interest" reads unclear. What do the authors mean by "intervention concept" and "outcome concept"? How could IR be considered an "intervention concept"? This needs to be revised.

Response: The section has been revised (lines 201 – 206)

Comment: If your Context is the whole African continent, you should at least mention this instead of stating "Africa" broadly, and also specify the nb of countries of "Africa" you will be investigating. Also, please note that the Canary Islands are part of Spain...

Response: The statement has been clarified as suggested (212 – 213)

Comment: Languages. In appendix 1, your search terms are exclusively provided in English. Why do you state that there will be no restrictions of language? By looking only into these common English-language databases, you will de facto primarily find publications in English language. Also, if you really aim to reach this ambitious goal, you will need to a) look into each non-English publication databases (and provide their names in the current protocol) and b) make sure to provide details on translation issues (what translating tools will be used? what quality checks?, etc).

Response: The language eligibility criterion has been revised (lines 245)

Comment: Exclusion criteria: on what basis would you remove items "evaluating implementation outcomes using specific IR approaches"?

Response: The statement has been revised for clarity (Lines 217 – 220)

Comment: Unclear whether the authors will perform a quality assessment of the selected papers.

Response: This has been clarified (lines 297 – 299)

Comment: Appendix 1: why would you look for "evidence-based medicine" papers? This is not coherent with your research focus. There are also a sheer number of keywords that are missing from this provisional search strategy. Please refer to existing published systematic and scoping reviews on the topic of UHC.

Response: The search strategy has been revised accordingly.

Comment: Lastly, many tools are now freely available to researchers who perform literature reviews. I would strongly recommend the authors to use a common review extraction tool, such as Rayyan (<https://eur01.safelinks.protection.outlook.com/?url=https%3A%2F%2Frayyan.qcri.org%2Fwelcome&data=04%7C01%7Cnnjchu001%40myuct.ac.za%7C23c12ff9896942b5002008d875b89d40%7C92454335564e4ccfb0b024445b8c03f7%7C0%7C0%7C637388783062272068%7CUnknown%7CTWFpbGZsb3d8eyJWljiMC4wLjAwMDAiLCJQIjoiV2luMzliLCJBTiI6Ikl1haWwiLCJXVCi6Mn0%3D%7C1000&data=IUxDP4XwugbudqRgjpgd%2BGD2B6aWSPmd%2FKN24WMP7hL0%3D&reserved=0>) as it would ensure a more transparent and rigorous item screening and selection than Excel.

Response: We appreciate the suggestion of this potentially valuable tool. While we will use the Excel-based data extraction for the proposed review (given that we have found this much easy to use in previous reviews), we will consider trying out the suggested tool in future reviews.

Comment: Please remove the sections "Ethics and dissemination" and "Patient and public involvement" as these are not applicable to a scoping review protocol.

Response: We have removed the heading "Patient and public involvement", but retained that of "Ethics and dissemination" as it was required by the editor during the initial submission, and the heading features in recently published BMJ scoping review protocols.

Comment: Unclear why the PRISMA guidelines are provided at the end of the submission - especially since the authors do not apply the PRISMA-recommended PICO strategy.

Response: We have adapted and used the more appropriate PRISMA-ScR checklist for scoping review. Thank you.

Comment: While the level of English language is acceptable in general, I would suggest a thorough English proofreading done by a professional language editor.

Response: We appreciate the advice. We have proof-read and enhanced the manuscript for better coherence.

Reviewer 2

Comment: This scoping review protocol by Nnaji and colleagues addresses an applied-research question of great importance to health policy and practice in Africa. The review will attempt to identify and characterize health implementation research initiatives in African countries. I do not have any major concerns with the protocol, but I offer a few comments and suggestions for the authors' consideration:

Response: We are thankful to the reviewer for appreciating the importance of our review.

Comment: p10 (line 27): some experts think that the UHC cube does not adequately address quality of care. In this conceptual framework (and in the study inclusion/exclusion criteria and charting approach), I think the authors could distinguish between "population coverage" (the name of the dimension on the UHC cube), which is mostly about aggregate access, and quality, which is a modifier of individual health outcomes. My read of the literature suggests that most implementation research studies tend to focus on either improving access (expanding coverage to marginalized clients) or improving the quality of care received by those already in the system. Of course, there are some overlaps. Ng and colleagues (PLoS Medicine, 2014) talk about "effective coverage" as the combination of access (utilization) and quality, and I think it would be helpful to use this term or at least to disentangle quality improvement from population coverage.

Response: We thank the reviewer for this insightful suggestion. We have revised the review's conceptual framework (lines 165 – 172) and have described the UHC related outcome under inclusion criteria (lines 209 – 211).

Comment: p11 (stage 1): in light of the above, I might suggest re-wording or re-structuring the research objectives a bit. The 4 research questions do not align well with the conceptual framework (UHC cube); they introduce new constructs (eg, resilience/sustainability; barriers to uptake of research) and do not speak at all to two of the three UHC cube dimensions: financial protection and the scope of the UHC benefits package (ie, services covered under UHC).

Response: We have revised the research questions accordingly objectives section (lines 159 – 161).

Comment: p12 (inclusion criteria): again, I suggest being more clear about the UHC outcomes of interest. Lines 20-24 mention health service coverage (not sure if this refers to population coverage or scope of benefits package), access (including utilization and quality together - see above comments), and financial protection... but not much about resiliency, evidence-to-policy, etc. I suggest a modification of Figure 1 that shows the relationships between the study objectives and the dimensions of the cube and the outcomes of interest, to make the whole strategy feel more coherent.

Response: We have revised the UHC outcomes (Line 209 – 211)

Comment: p12 (exclusion criteria): the authors say they will exclude "outcomes that are not UHC-related." Some additional explanation is warranted here. If the scope of this review is only studies that look at general health systems strengthening to improve overall access, financial protection, etc, then I think the search will turn up with very little. However, there could be a wide range of disease- or program-specific studies that speak to UHC outcomes like access and financial protection without explicitly using the term UHC or looking at the whole health system. For example, a paper may use a quasi-experimental approach (and robust implementation science methods) to evaluate the impact of

a new HIV care delivery model on ART uptake and out-of-pocket costs. What to do about this sort of study? It might be useful for the authors to cite a few exemplar studies (eg, at the end of this section at the top of p13) to give the reader a better sense for what sort of implementation research studies the authors hope to pick up (or not) with the review.

Response: We are grateful for this insightful comment and suggestion. We agree that given the system-wide implications of UHC and its outcomes, it will be prudent to keep an open mind in searching for relevant publications rather than restricting the UHC inclusion criteria. Hence, we have redacted that aspect of the exclusion criteria.

Comment: p13 (line 30): minor point: why search databases "from inception"? The modern UHC movement only goes back to about 2005, and the field of implementation research is not much older. I worry about clouding the review with a bunch of studies of vertical programs from the 1980s/90s that have less relevance in today's time.

Response: We appreciate this valuable feedback. We have revised the search timeline as advised.

Comment: p14 (line 37): another minor point: I might explicitly include protocols for research studies in the inclusion criteria and in the extraction/charting process. I would guess that there are a number of large-scale multi-year studies that are ongoing but not published (except in protocol form).

Response: We are restricting the inclusion to actual implementation research conducted, hence will exclude study protocols (lines 219 – 220).

Comment: p15 (stage 7): the authors could consider providing a few examples of how they intend to visualize and interpret the charted evidence. Maybe mention some relevant disaggregations or ways of summarizing the evidence?

Response: We have discussed this in lines 294 – 297.

Comment: p16 (lines 7-9): I wonder whether stakeholders should be involved in the identification of grey literature at the study identification phase (lit search) rather than at the dissemination phase?

Given the broad objective of this research, it may be worthwhile to have a multidisciplinary, multinational advisory committee for this work that could be engaged from beginning to end.

Response: We have clarified our stakeholder engagement approach (lines 304 – 308)

VERSION 2 – REVIEW

REVIEWER	Lara Gautier Université de Montréal, Canada
REVIEW RETURNED	10-Dec-2020

GENERAL COMMENTS	Thanks for this revised version, the content of the manuscript has improved. I appreciate the use of Ridde et al's taxonomy. However, the research objective is still unclear and needs to be revised. I still do not understand the link between implementation research and UHC promotion, and I do not understand what would be the point of 'identifying and characterising the nature, facilitators and barriers of implementation research initiatives used for promoting UHC in the Africa region'. What do you mean by 'implementation research initiatives'? do you mean UHC projects being evaluated using implementation research? Please clarify. I think what you ought to do is simply a scoping review of IR papers on UHC in Africa, which you will characterise according to the aforementioned taxonomy. You will be able to identify research
--

	gaps too. This focus will be much clearer and allow for a much more insightful review. Thus I still do not understand the following research question: 'What are the contextual facilitators and barriers to the uptake and sustainability of implementation research for promoting UHC in Africa?'. Most importantly, I do not see how you will yield information on this based on the present scoping review protocol, which uses strategies that enable to document implementation research on UHC in the African continent, rather than 'implementation research for promoting UHC'. Again, the relationship between IR and UHC advocacy is unclear to me, and I am not sure you should elaborate on this. Also - if you are interested in implementation research, why do you mention 'UHC outcomes' in Stage 8 section? Many synonyms are still missing from the search strategy. Think about process evaluation, etc. Please ask a librarian to help you with this. Some sections of the search strategy also appear to lack coherence. For instance, it is not clear to me why you have ["evidence-based healthcare"[Title/Abstract] OR "evidence-based health care"[Title/Abstract] OR "evidence-informed decision making"[Title/Abstract] OR "evidence-informed healthcare decision making"[Title/Abstract]] in your search strategy. What do you aim to achieve with this addition? It may exclude a lot of papers. Also not sure about 'knowledge translation' and its synonyms. Why do you add this? I did not see anywhere that you focus on knowledge translation of UHC-relation IR findings? Please ensure coherence.
--	--

REVIEWER	David Watkins University of Washington
REVIEW RETURNED	30-Nov-2020

GENERAL COMMENTS	This manuscript is much improved upon the original version. My only remaining comment is that the manuscript does not quite describe how the data will be used to inform the 2 research questions on p9 (stage 1). In particular, the extraction and charting process (stages 6-7) does not describe how the authors will extract and chart/synthesize "contextual facilitators and barriers to the uptake and sustainability of implementation research" (RQ 2) from individual studies. RQ 1 is also vaguely written but could at least be answered based on what is described in stage 7.
---

VERSION 2 – AUTHOR RESPONSE

Reviewer 1:

Comments: Thanks for this revised version, the content of the manuscript has improved. I appreciate the use of Ridde et al's taxonomy.

Response: We thank the reviewer for appreciating our revision of the manuscript.

Comment: However, the research objective is still unclear and needs to be revised.

Response: We have revised the objectives and research questions for better clarity (Lines 158 – 160 and 189 – 192).

Comment: I still do not understand the link between implementation research and UHC promotion, and I do not understand what would be the point of 'identifying and characterising the nature, facilitators and barriers of implementation research initiatives used for promoting UHC in the Africa region'.

Response: We have revised the objectives and research questions for better clarity. We have now focused on the use of implementation research in the evaluation/assessment of UHC-related programmes, rather than for promotion of UHC. (Lines 158 – 160 and 189 – 192).

Comment: What do you mean by 'implementation research initiatives'? do you mean UHC projects being evaluated using implementation research? Please clarify. I think what you ought to do is simply a scoping review of IR papers on UHC in Africa, which you will characterise according to the aforementioned taxonomy. You will be able to identify research gaps too. This focus will be much clearer and allow for a much more insightful review.

Response: We have revised the objectives and research questions for better clarity. We have now focused on the use of implementation research in the evaluation/assessment of UHC-related programmes, rather than the promotion of UHC. (Lines 158 – 160 and 189 – 192).

Comment: Thus I still do not understand the following research question: 'What are the contextual facilitators and barriers to the uptake and sustainability of implementation research for promoting UHC in Africa?'. Most importantly, I do not see how you will yield information on this based on the present scoping review protocol, which uses strategies that enable to document implementation research on UHC in the African continent, rather than 'implementation research for promoting UHC'. Again, the relationship between IR and UHC advocacy is unclear to me, and I am not sure you should elaborate on this.

Response: We have revised the objectives and research questions to focus on the nature, facilitators and barriers of the use of implementation research for evaluating/assessing of UHC-related programmes/interventions, rather than for promoting of UHC. (Lines 158 – 160 and 189 – 192).

Comment: Also - if you are interested in implementation research, why do you mention 'UHC outcomes' in Stage 8 section?

Response: We have corrected this to rather focus on the type of UHC-related programme or intervention involved, rather than UHC outcomes (Lines 297-298)

Comment: Many synonyms are still missing from the search strategy. Think about process evaluation, etc. Please ask a librarian to help you with this.

Response: We have revised the search strategy (See revised Appendix 1)

Comment: Some sections of the search strategy also appear to lack coherence. For instance, it is not clear to me why you have ["evidence-based healthcare"[Title/Abstract] OR "evidence-based health care"[Title/Abstract] OR "evidence-informed decision making"[Title/Abstract] OR "evidence-informed healthcare decision making"[Title/Abstract]] in your search strategy. What do you aim to achieve with this addition? It may exclude a lot of papers.

Also not sure about 'knowledge translation' and its synonyms. Why do you add this? I did not see anywhere that you focus on knowledge translation of UHC-relation IR findings?

Response: The search strategy has been revised accordingly (See revised Appendix 1)

Reviewer 2:

Comment: This manuscript is much improved upon the original version.

Response: We thank the reviewer for appreciating the improved draft of the revised manuscript.

Comment: My only remaining comment is that the manuscript does not quite describe how the data will be used to inform the 2 research questions on p9 (stage 1).

Response: We have revised the research questions for better clarity and conformity with the review's objective (Lines 189 – 192).

Comment: In particular, the extraction and charting process (stages 6-7) does not describe how the authors will extract and chart/synthesize "contextual facilitators and barriers to the uptake and sustainability of implementation research" (RQ 2) from individual studies.

Response: We appreciate this comment. We have described how we plan to extract, chart and synthesis the data related to facilitators and barriers (lines 276, 287-288 and 300-301).

Comment: RQ 1 is also vaguely written but could at least be answered based on what is described in stage 7.

Response: We have revised the research questions for better clarity (Lines 189 – 192).

VERSION 3 – REVIEW

REVIEWER	David Watkins University of Washington, USA
REVIEW RETURNED	26-Jan-2021

GENERAL COMMENTS	The minor revisions made by the authors have improved the manuscript. I have no further comments or concerns.
---